# Vocabulary embeddings organize linguistic structure early in language model training

## Abstract

Large language models (LLMs) work by manipulating the geometry of input embedding vectors over multiple layers. Here, we ask: how are the input vocabulary representations of language models structured, and how and when does this structure evolve over training? To answer this question, we use representational similarity analysis, running a suite of experiments that correlate the geometric structure of the input embeddings and output embeddings of two open-source models (Pythia 12B and OLMo 7B) with semantic, syntactic, and frequency-based metrics over the course of training. Our key findings are as follows: 1) During training, the vocabulary embedding geometry quickly converges to high correlations with a suite of semantic and syntactic features; 2) Embeddings of high-frequency and function words (e.g., "the," "of") converge to their final vectors faster than lexical and low-frequency words, which retain some alignment with the bias in their random initializations. These findings help map the dynamic trajectory by which input embeddings organize around linguistic structure, revealing distinct roles for word frequency and function. Our findings motivate a deeper study of how the evolution of vocabulary geometry may facilitate specific capability gains during model training.

## 1 Introduction

Token embeddings are the input vectors to transformer language models. The information that differentiates one input from another, and spurs the diverse and complex processing in large language models, all originates in the vector space of the token embeddings. Understanding the structure of vocabulary embedding representation is therefore a fundamental step in the effort to trace and interpret the internal mechanisms of language models. In this paper, we analyze the representational space of the token embeddings of 153 Pythia 12-billion checkpoints (Biderman et al., 2023) and 186 OLMo 7-billion checkpoints (Groeneveld et al., 2024), and analyze how the representational relationships in the vocabulary matrix form over the course of training. We track how vocabulary embeddings evolve over training to reflect relationships in 1) semantics, 2) syntax, and 3) word frequency, and how the organizational principles of syntactic class and word frequency affect the convergence of the embedding vectors over training.

Our main analytical approach is Representational Similarity Analysis (RSA; Kriegeskorte et al., 2008; Nili et al., 2014), which measures how correlated the pairwise distance relationships between two representations are. The primary benefit of RSA is that it provides a framework for comparing very different artifacts as long as they are acting on the same stimuli (in our case, the stimuli are English words), and can be defined in terms of a distance metric. RSA lets us compare continuous model vector representations with annotations of the English vocabulary that are in an unrelated medium, such as human word similarity judgments and part-of-speech annotations (Section 4), or word frequency counts (Section 5). Key to our approach is that the English vocabulary is an extremely well-annotated artifact, providing diverse metadata to contextualize the vocabulary space that is the focus of our analyses.

We operationalize RSA in two different ways to understand different aspects of the development of vocabulary structure over training. First, we use **Hypothesis-Driven RSA** (henceforth, "hypothesis RSA"; Kriegeskorte et al., 2008; Jozwik et al., 2017; Zhang et al., 2017; Groen et al., 2018; Dong et al., 2022; Goddard & Mullen, 2024) to determine when during training and to what degree model

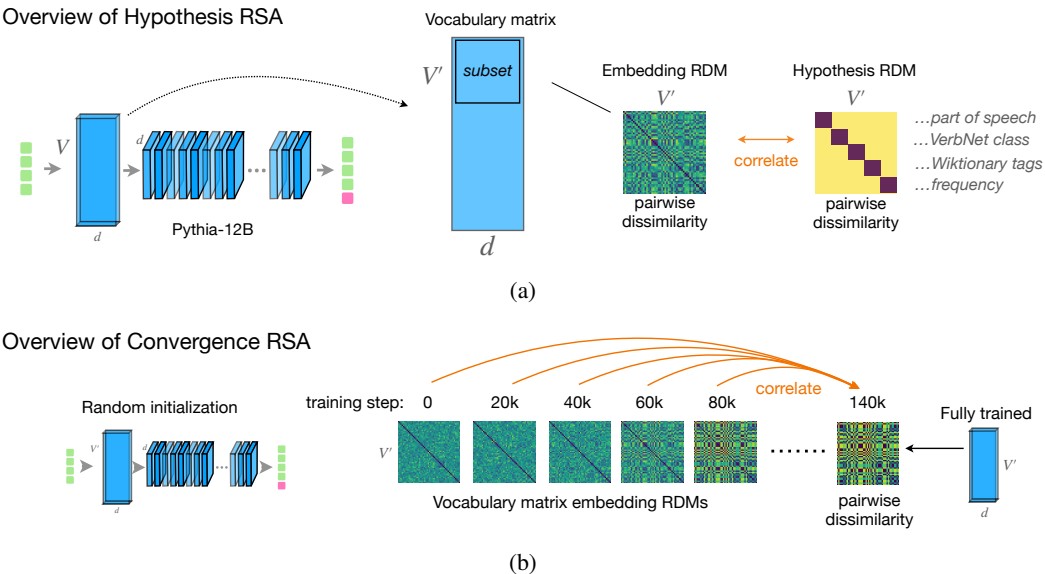

Figure 1: A schematic illustrating our two uses of RSA. In Hypothesis RSA, we take the vocabulary matrix, correlating models to annotated hypotheses, and tracking the convergence of different classes of words

representations encode relationships derived from external annotation sources, such as human word similarity judgments or part-of-speech groupings. Second, we examine how significant token groupings (such as tokens of the same part of speech) evolve their intra-group relationships during training to reach their final configuration in the fully trained model, a method we call **Convergence RSA**. Convergence RSA lets us see how different subsets of the vocabulary change and stabilize during training. This approach is similar in spirit to Representational Trajectory Analysis (Kallmayer et al., 2020), but we study how a given representation evolves throughout training, rather than over the course of model layers.

**In Experiment 1, we trace the development of semantic and syntactic structure over training** (Section 4). Correlating the vocabulary matrix to datasets of human word similarity judgments (SimLex and WordSim: Hill et al., 2015; Finkelstein et al., 2001; Agirre et al., 2009), as well as syntactic measures (such as part-of-speech or verb class). We find that model vocabulary structure peaks in correlation with these structures at around 15% of training.

**In Experiment 2, we trace the effect of word frequency over vocabulary training** (Section 5). We show that frequent words stabilize their representations more rapidly, while less frequent words maintain correlations with relationships present in the random initialization. In contrast, the most frequent words completely shed these initialization biases. Moreover, we see that throughout training, the vocabulary matrix progressively develops to represent frequency rank relationships (where words with similar frequency rankings become closer). This reveals how vocabulary representations continue to evolve even after semantic and syntactic measures stabilize early in training.

**In Experiment 3, we run a series of analyses to understand what changes after linguistic features stabilize** (Section 6). We find that embeddings continue changing rapidly, but that the correlations between input and output embeddings decreases after linguistic stabilization. We also find a remarkably consistent qualitative result: among all vocabulary words, those experiencing the greatest changes in the last 85% of training are rare (often technical) words moving closer to their morphological inflections.

## 2 BACKGROUND AND RELATED WORK

### 2.1 STRUCTURE IN THE LEXICON

How much information is encoded in a word, and how much in how words are put together? How can we distinguish between lexical representation and grammatical structure, and where does one

end and the other begin? These questions are fundamental to understanding how the human and machine language systems operate. Here, we give a brief overview of the complex and meaningful structures that exist in the lexicon.

**Structure in the vocabulary** While many syntactic analyses describe abstract grammatical systems like recursion that operate independently from a speaker's lexical semantic knowledge (see Hauser et al. (2002); Pinker (1998); Chomsky (1994) for some notable examples of this "words and rules" approach), several theoretical frameworks highlight the rich syntactic information embedded within the lexicon and its interface with grammar. One influential alternative perspective is frame semantics: the idea that words contain information about which kinds of words can go with them, or what frames they can appear in (Fillmore et al., 2006; Baker et al., 1998; Levin, 1993; Kipper et al., 2008). Pustejovsky (1998) proposes that the lexicon is highly generative, accounting for subtle related meaning variations (such as "fast" meaning going quickly, or, lasting a brief time) and creative extensions of word meaning (as in "Corvetted across the USA") (Clark & Clark, 1979). Lastly, diverse work explores the idea of the syntax-lexicon continuum (Croft, 2001; 2020; Tomasello, 2005; Jackendoff, 2011; Goldberg, 1995): that there is no clear separation between single words, semi-fossilized structures like "not only... but also", and more general grammatical patterns like the passive voice. Following these analytical ideas, we examine linguistic structure in the vocabulary, contributing to a program of interpretability that does not a priori limit what information is in what part of the model.

**Frequency in the vocabulary** One fundamental fact across human languages is that their vocabularies follow a Zipfian (or, power-law) distribution (Zipf, 1936; Mandelbrot, 1953), where there are few words that make up the bulk of words in any given sentence ("the", "of"), and many words with very low frequency ("hedgehog", "popsicle') (for a comprehensive review of Zipfian vocabulary distributions and related theories, see Piantadosi (2014)). The power-law distribution of the vocabulary is one of its basic structuring principles, and here we investigate how the vocabulary matrix interfaces with it in Section 5.

## 2.2 THE EMBEDDING MATRIX IN LLMS

Most transformer language models encode word meaning in a word embedding matrix of dimension $(V \times d)$, where $V$ is the size of the tokenizer vocabulary, which splits most text into units that roughly represent meaningful words (in many language models $V$ is around 50-100 thousand), and $d$ is the model dimension. Each token in the tokenizer is assigned a vector in the embedding matrix that represents it, and a row of the embedding matrix cannot receive any gradient from backpropagation unless the specific token has appeared in the current batch (some exceptions to this dominant tokenization paradigm include Clark et al. (2022), Xue et al. (2022), Ahia et al. (2024), Kallini et al. (2025), Huang et al. (2023), and Feher et al. (2024)). In most standard transformer models, all subsequent activations are derived by manipulating and establishing relationships between the initial vectors from the embedding matrix.

While substantial interpretability research has focused on model activations in later model layers, and early foundational interpretability work focused on understanding the geometric structure of static word embeddings like word2vec (see Mikolov et al. (2013); Pennington et al. (2014); Ethayarajh et al. (2019); Hashimoto et al. (2016); Allen & Hospedales (2019)), and a line of work looks at more general model training dynamics Chen et al. (2023a); Tirumala et al. (2022), research examining the embedding matrix of language models from the training perspective remains comparatively sparse. Past work has shown complex lexical representations even in models with insufficiently expressive tokenizers (Feucht et al., 2024), tracked the evolution of grammatical interpretations from word embeddings to later layers (Papadimitriou et al., 2022), analyzed how LM gradients project onto the vocabulary matrix (Katz et al., 2024), and examined how rare undertrained tokens affect model behavior (Land & Bartolo, 2024). Others have also explored how vocabulary re-learning influences cross-lingual transfer (Wu et al., 2023; Patil et al., 2022; Chronopoulou et al., 2020; Chen et al., 2023b), the effects of using vastly different sizes of vocabulary matrices (Liang et al., 2023; Schmidt et al., 2024), and the role of vocabulary in scaling language models (Huang et al., 2025; Wies et al., 2021). Our work introduces a comprehensive framework for representational analysis that lets us systematically analyze the encoded linguistic structure in embeddings across training.

### 2.3 Hypothesis-driven Representational Similarity Analysis

Representational Similarity Analysis (RSA) quantifies how stimuli are encoded by comparing their evoked response patterns (Kriegeskorte et al., 2008; Nili et al., 2014). By constructing Representational Dissimilarity Matrices (RDMs) from pairwise comparisons of multivariate responses (from fMRI, neural recordings, or model activations (Lepori & McCoy, 2020)), RSA enables correlation between different representational systems containing different numbers of features. In hypothesis-driven RSA, theoretical accounts are translated into interpretable RDMs for direct comparison with empirical data, allowing researchers to test whether specific regions or networks organize information by semantic categories or other metrics (Jozwik et al., 2017; Groen et al., 2018). For instance, animacy RDMs can help determine if visual cortex encodes animate/inanimate distinctions (Nili et al., 2014). We apply this approach to track how vocabulary representations in language models evolve throughout training, examining both their alignment with linguistic features and the developmental trajectories of semantically and syntactically defined word groups.

## 3 Methods

We next outline the key methodological choices underlying all experiments, with dataset details and experiment-specific implementations described in methods sections corresponding to each analysis.

**Models and embedding matrices** We use the 153 released training checkpoints of the 12-billion-parameter Pythia-12B model (Biderman et al., 2023), and 186 released checkpoints of the OLMo-7B model (Groeneveld et al., 2024) (this is every third checkpoint of the 558 released checkpoints, due to resource constraints). In all, we run the analyses outlined in this paper on a **total of 339 separate language model instances**. We run experiments on the input and output embeddings of the models, which are not tied in both Pythia and OLMo (i.e., they are separate embeddings of the lexicons). Though understanding the effect of tying the input and output embeddings would be interesting (Press & Wolf, 2017; Bertolotti & Cazzola, 2024), we are not aware of any large, modern models that open-source their training checkpoints and have weight-tying. For brevity, we only show Pythia input embeddings results in the figures in the main text. Results for OLMo can be found in Appendix A, and results for output embeddings in Appendix B. Both sets of findings are discussed in the main text where relevant.

The Pythia and OLMo models share an identical tokenizer vocabulary (and embedding matrix length) of 50,688, of which 28,000 are full words and word starts, with the rest being subword continuation tokens. Throughout our analyses, we focus on full word tokens: tokens that are in exact correspondence with the words of our annotated datasets. Understanding the dynamics and geometry of subword tokens remains an interesting avenue for future work.

**Distance in model embedding space** Throughout the analyses in this paper, we use Spearman distance between embedding vectors as a distance measure, which indicates whether the dimensions of the two vectors have the same ranking, regardless of magnitude. This follows the best pratices suggested by Zhelezniak et al. (2019), and the results of Timkey & van Schijndel (2021), who found that oversized dimensions with little causal influence on model behavior can distort other representational distance measures. To ensure our choice of measure does not substantially affect results, we replicated the semantic experiments from Experiment 1 using both Euclidean and cosine distance, obtaining nearly identical outcomes (see Appendix C, Figure 9).

**Hypothesis RSA: correlation with annotated features (Figure 1a)** To run our hypothesis-driven analyses, we take the vocabulary matrix of dimension $V \times d$ and create a pairwise distance matrix (a representational dissimilarity matrix, RDM) $D_{\text{model}} \in \mathbb{R}^{V \times V}$, using the Spearman distance between the embeddings corresponding to each pair of words. Then, for each annotated hypothesis dataset, we construct a dissimilarity matrix over the subset of words appearing in the dataset, yielding $D_{\text{hypothesis}} \in \mathbb{R}^{V' \times V'}$, where $V' < V$. To compare with the model, we extract the corresponding $V'$ rows and columns from $D_{\text{model}}$, producing $D'\text{model} \in \mathbb{R}^{V' \times V'}$. We then compute Kendall's $\tau$ correlation between the vectorized lower triangular entries in $D_{\text{hypothesis}}$ and $D'_{\text{model}}$, which quantifies the representational similarity between the vocabulary matrix and the annotated hypothesis. This procedure is repeated for every model checkpoint to track how similarity evolves over training.

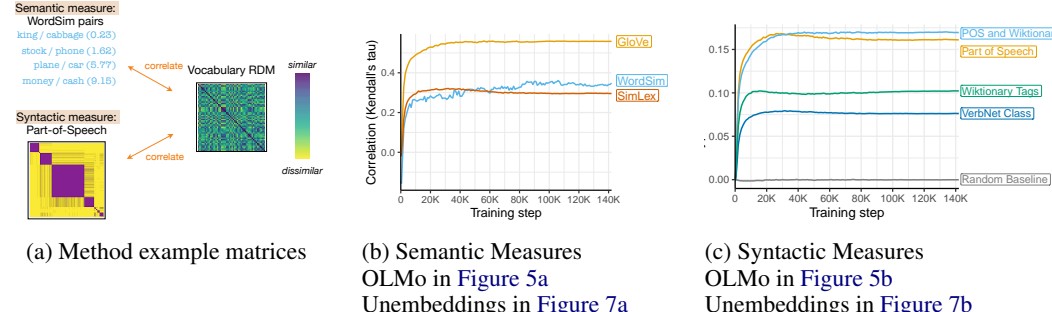

(a) Method example matrices

(b) Semantic Measures
OLMo in Figure 5a
Unembeddings in Figure 7a

(c) Syntactic Measures
OLMo in Figure 5b
Unembeddings in Figure 7b

Figure 2: **Experiment 1: correlation with semantic and syntactic similarity measures**. We compare the distance relationships in the model vocabulary embedding with the distances in different measures of semantic and syntactic similarity. **(b)** Model embeddings come to represent semantic similarities quickly, with correlations converging quite early in training. **(c)** Model embeddings correlate with syntactic structural RDMs early in training, peaking then plateauing. Note that the y-axes differ across the two plots: each syntactic hypothesis captures a relatively simple relation compared to the more complex semantic relationships, which likely explains the lower overall correlation plateaus.

**Convergence RSA: correlation with final training checkpoint (Figure 1b)**  In Convergence RSA, instead of comparing the structure of the vocabulary matrix against external hypotheses, we measure how the representational geometry at each checkpoint correlates with that of the final trained model. For a given subset of tokens $V'$, we create a representational dissimilarity matrix $D'_{\mathrm{model},i} \in \mathbb{R}^{V' \times V'}$ at training step $i$ using pairwise Spearman distances. We then compute Kendall's $\tau$ correlation between $D'_{\mathrm{model},i}$ and $D'_{\mathrm{model,final}}$, yielding a measure of similarity between the representation at step $i$ and the fully-trained representation. This approach allows us to track how different subsets of the vocabulary converge to their final representational structure over the course of training.

## 4 EXPERIMENT 1: HYPOTHESIS RSA WITH SEMANTIC AND SYNTACTIC MEASURES

Our first investigation focuses on understanding when model token embeddings represent relationships of meaning and language structure, using annotated semantic and syntactic measures .

### 4.1 METHODS: SEMANTIC AND SYNTACTIC MEASURES

**Semantic Similarity Measures**  For the semantic similarity measures, we use word-pair similarity annotations. Since semantic similarity measures contain sparse human annotations between words (i.e, not every possible word pair has an annotation associated with it), we cannot correlate with the full vocabulary RDM, and instead report correlations with the vector corresponding to the embedding distances between the annotated pairs. The datasets we use are:

**1) the WordSim-353 Dataset** (Finkelstein et al., 2001) contains 353 word pairs with similarity ratings averaged across 13-16 human subjects.
**2) The SimLex-999 dataset** (Hill et al., 2015) contains 999 word pairs, covering nouns, adjectives, and verbs, annotated by 500 subjects. For both WordSim and SimLex, our reported correlations include only those word pairs where both words exist in the Pythia/OLMo tokenizer.
**3) GloVe word embedding vectors** GloVe vectors (Pennington et al., 2014) represent word meaning in a high-dimensional space, encoding important semantic relationships like analogies linearly. We take the set of 983 word pairs from SimLex that's common across the GloVe vocabulary and the Pythia/OLMo tokenizer, so the GloVe results are comparable with the word similarity datasets.

**Syntactic Similarity Measures**  For the syntactic similarity measures, we take syntactic lexicon annotations and make hypothesis RDMs out of them. For example, for part-of-speech annotations, the hypothesis distance matrix is that words with the same part of speech have distance 0 and words

with different parts of speech have distance 1. We emphasize that each hypothesis is relatively simple, so we do not expect high absolute correlation values, as those would imply that embeddings encode only a single basic relation like part of speech. Instead, the correlations are best interpreted in a comparative sense, either across hypotheses or across training stages, rather than as absolute measures of encoding strength. We detail our syntactic hypothesis RDMs below:

**1) Part of speech RDM** The first syntactic feature that we test is part of speech: the syntactic classes like verbs and nouns. We use the human-corrected EWT Universal Dependncies treebank parses (Silveira et al., 2014; de Marneffe et al., 2021; Nivre et al., 2020), and if a word appears as a part of speech at least 5 times it receives distance 0 to all other words with that part of speech.
**2) Wiktionary Tags RDM** For a fine-grained set of features, extract the Wiktionary tags of words (syntactic tags such as *transitive* or regional/register tags such as *informal*), and create an RDM where words that share a tag have distance 0 (we use Ylonen, 2022, Wiktextract). We subsampled 3,000 words (of the 21,500 possible words) for constructing our RDM in order to make pairwise distance calculations computationally feasible. We ran thhe experiment 5 times and saw only small variations (standard deviation of $< 0.23$ throughout every training checkpoint).
**3) VerbNet classes RDM** We create a hypothesis RDM where verbs of the same VerbNet verb class have distance 0. VerbNet (Korhonen & Briscoe, 2004; Kipper et al., 2006; 2008) is a comprehensive resource carefully annotated by experts that categorizes verbs into verb classes (Levin, 1993): sets of verbs that take the same verb arguments and the same alternations.
**4) Random Baseline** As a conservative control, we assess how model embeddings correlate with a 'grouping' metric that lacks any syntactic information. This baseline consists of a hypothesis RDM with the same number of classes as VerbNet and the same number of words per class, but with randomly sampled tokens in each class. The correlation with this RDM gives a baseline for how much token embeddings correlate with distances that correspond to consistent groupings of words, regardless of whether these groupings are in some way meaningful.
**5) Part of Speech *and* Wiktionary RDM** Lastly, we evaluate model correlations to a dissimilarity matrix that combines information from part of speech and Wiktionary. In this hypothesis RDM, distances reflect graded overlap: two words receive a distance of 0 if they share both tags, 0.25 if they share only part of speech, and 0.5 if they share only a Wiktionary tag. This scheme heuristically encodes the intuition that joint agreement is stronger than either source alone.

## 4.2 RESULTS

**Semantic information arises early in training** We present our results in Figure 2. Overall, Figure 2b shows that the token embeddings of the model converge to their approximate final correlation relatively quickly — within 10,000 steps for correlation with SimLex distances. This indicates that the semantic structure of vocabulary representations is established very early in training. Results for OLMo and for output embeddings are very similar. Results for OLMo are very similar, suggesting that this early emergence of semantic structure is not specific to a particular architecture or training setup but may be a more general property of LMs. We also find that output embeddings behave similarly, indicating that the semantic organization is shared across both input and output vocabularies. All together, our findings imply that the rapid emergence of semantic structure is both model-agnostic and robust across different embedding roles.

Interestingly, we also observe that model embeddings converge to encode similarity much more than relatedness (see Appendix D, Figure 10). The Similarity and Relatedness splits of WordSim (Agirre et al., 2009) disambiguate similarity (words like 'car' and 'truck') and relatedness (words like 'car' and 'road', which are related but very dissimilar objects). We find that at the final checkpoint, model embeddings are significantly more correlated with the WordSim-Similarity split (Pythia correlation 0.56) than with the WordSim-Relatedness split (Pythia correlation 0.21), a distinction which static embeddings fail to make (the anaalogous GloVe difference is 0.59/0.50).

**Syntactic organization of the vocabulary peaks early in training** The syntactic results in Figure 2c show that model embeddings converge to their final correlations with our syntactic organization hypotheses early in training. In fact, for all three of our syntactic RDMs (part of speech, Wiktionary tags, and VerbNet classes), the correlation peaks early and then stabilizes to a lower value than the peak. We see similar results in OLMo, with one key difference: though the POS embeddings have a sharp elbow at a similar point in training, and at a similar correlation value to Pythia,

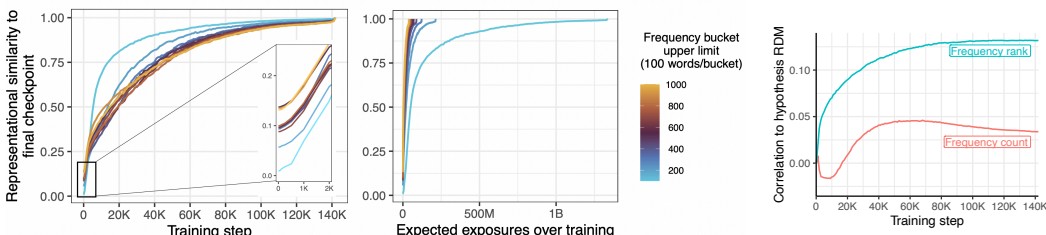

(a) Convergence RSA by frequency buckets
OLMo in Figure 6, Unembeddings in Figure 8

(b) Hypothesis RSA to frequency

Figure 3: **Experiment 2: The effect of frequency on the vocabulary (a)** Convergence of different frequency buckets (**left**): Frequent words (blue) converge to their final representations faster than infrequent words (orange; see **inset**). Less frequent words have correlated representational structures between their random initializations and their final checkpoints (**right**). The same figure, but with the x-axis rescaled independently for each line to reflect the expected number of times the model has seen the words in each frequency bucket, showcasing how frequent words evolve much slower per update. **(b)** Vocabulary embedding RDM correlations with frequency hypothesis distance matrices. During training, distances in vocabulary space gradually align with differences in frequency rank, though the relationship to raw frequency counts is non-monotonic.

they then continue rising. This suggests an interesting direction for future work: understanding why OLMo attains stronger POS correlations than Pythia, even when other results are nearly identical. One notable exception to the Pythia early-peak pattern is also the best-correlating hypothesis: the combination of part-of-speech and Wiktionary tags. Correlation to this joint hypothesis RDM grows more monotonically and stabilizes slightly later than any of the other hypotheses. This points to the possibility that the relationships that arise in model embeddings over training are better explained by nontrivial combinations of linguistic features, and understanding these interactions is an interesting avenue for future investigation in interpretability. In sum, we find that semantic structure emerges rapidly and achieves the strongest correlations overall, while syntactic organization peaks early at lower levels, with the best performance coming from combined linguistic feature hypotheses.

## 5 EXPERIMENT 2: THE EFFECT OF FREQUENCY

A fundamental and far-reaching aspect of language is that the vocabulary of human languages follows a power-law distribution. Here, we examine how the vocabulary matrix encodes the structural effects of vocabulary frequency over training.

### 5.1 METHODS

**Convergence RSA by frequency buckets** As in our part-of-speech analysis, we test how quickly different splits of the vocabulary converge to the final representations at the end of training. In order to test the effect of frequency, we split the top 1000 words in the vocabulary into 10 frequency buckets, and run Convergence RSA for each bucket, as illustrated in Figure 1b.

**Hypothesis RSA: correlation to frequency rank and frequency count** We also use frequency measures to construct two hypothesis RDMs: the Frequency-Rank and the Frequency-Count dissimilarity matrices. In the Frequency-Rank RDM, the distance between two words is how far apart they are in the frequency rank: the most frequent word and the 3rd-most-frequent word have a distance of 2. In the Frequency-Count RDM, the distance between two words is the difference in their frequency counts, and so they would have a distance of 2.3 billion, owing to the extreme disparity in occurrence counts between the most common and moderately common words in the distribution.

To make our frequency counts applicable across multiple models (as it is very expensive to count the frequency of words over a large corpus), we use a model-agnostic enhanced space tokenization regex and get frequency counts from C4, described in more detail in Appendix E.

### 5.2 RESULTS

**High frequency words converge quickly** As shown in Figure 3a, high-frequency words (blue) stabilize faster in training than low-frequency words (orange), with the top 100 most frequent words

showing significantly faster convergence than even the next bucket of 100 words. However, rescaling the x-axis by expected exposures per bucket reveals a different picture: high-frequency words change more slowly with each exposure, while low-frequency words shift more per occurrence and converge quickly despite fewer total exposures. Our results expose a dual effect of frequency: while high-frequency words stabilize earlier in absolute training time due to their frequent occurrence, each individual exposure produces smaller changes to their embedding vectors compared to low-frequency words. We show a related effect in Figure 11, Appendix F, where function words (grammatical like "the", which are also generally higher-frequency) converge faster than lexical words (words with specific meanings like "cat"). Due to the high correlations of these features, we cannot know if an effect is due to frequency or function.

**Low-frequency words are correlated with their random initializations**   In the inset of Figure 3a, we see that most lower-frequency words converge to representational structures that maintain correlation with their random initialization patterns. Less frequent words preserve stronger connections to their initialization vectors even at training completion. Only the most common 100 words fully break from their random initialization, showing 0 correlation between final embeddings and the starting vectors. This suggests that high-frequency tokens receive sufficient gradient updates to completely reshape their embedding structure, while rare words are biased by artifacts of their random initialization. OLMo results show the same patterns.

**Tokens move towards their frequency rank gradually throughout training**   As shown in Figure 3b, when we construct a hypothesis RDM based on frequency *rank*, we observe a gradual, monotonic increase in correlation across training. This indicates that the embedding matrix continues to evolve, with tokens progressively clustering according to their relative frequency positions (e.g., frequent vs. rare words). In contrast, when we use raw frequency *counts*, convergence is highly non-monotonic, implying that absolute occurrence counts do not provide a stable organizing principle. Taken together, these results suggest that the model embeds vocabulary in a way that reflects relative frequency rank rather than raw frequency, producing a more robust representational structure that strengthens steadily over training.

# 6   EXPERIMENT 3: HOW DO EMBEDDINGS CHANGE AFTER LINGUISTIC FEATURES STABILIZE?

As Experiment 1 shows, many key linguistic features reach their peak correlations very early in training. This raises a central question: after roughly 15% of training, do embeddings continue to change, or do they remain largely static? If they were static, the absence of further change in linguistic features would be expected. We next present three analyses demonstrating that embeddings do, in fact, continue to evolve well beyond the point at which linguistic features appear to stabilize.

**Embeddings keep changing after linguistic features stabilize**   In Figure 4a, we plot the average raw distance between 1,000 randomly sampled token embeddings at each checkpoint and their corresponding embeddings at the final training step. Unlike our RSA analyses, this measure does not capture relational geometry between tokens, but simply how far individual embeddings have moved in absolute terms. The results show that embeddings continue to shift substantially even after 20,000 steps, well beyond the point where correlations with linguistic features appear to have stabilized.

**Correlation between input and output embeddings peaks with linguistic representations**   In Figure 4b, we plot the correlation between input and output embeddings. Although Experiments 1 and 2 showed that both sets of embeddings correlate similarly with linguistic features, this does not imply that they are correlated with each other. Indeed, we find relatively low RSA correlation between their representational spaces overall. Notably, however, a clear peak appears at roughly 15% of training—the same point at which correlations with linguistic features peak. When bucketing by frequency, we also find a strong effect: high-frequency words (light blue) show substantially stronger correlations between input and output embeddings than low-frequency words.

**Qualitative analysis: The words that get closest after stabilization are inflections of rare nouns**
For a qualitative investigation of which tokens change the most after the first 20,000 steps, we calculate two $V \times V$ dissimilarity matrices: one at 20,000 steps and one at the final checkpoint

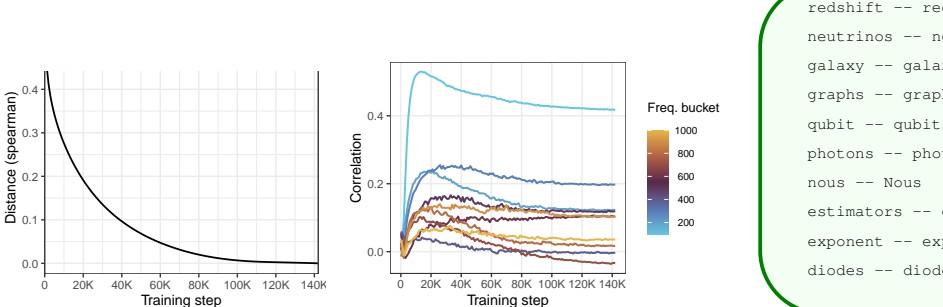

(a) Average distance between 1,000 random token embeddings and their embeddings at the final checkpoint

(b) Correlation between embeddings and unembeddings, by frequency bucket

(c) Top 10 words in Pythia that get closer between 20K and 142K steps. Full results in Appendix G, Figure 12 & Figure 13.

Figure 4: **Experiment 3** a series of analyses of what changes after linguistic feature stabilization.

(142,000 steps). We then take their difference, $D_{\text{diff}} = D_{\text{final}} - D_{\text{20K}}$, and examine the maximum and minimum values — that is, the word pairs that experience the greatest changes. We find that the largest decreases in distance (embeddings moving closer) are substantially greater in magnitude than the largest increases (embeddings moving farther), with a maximum difference of 0.476 vs. 0.176 in Spearman distance. The word pairs that have the greatest increase in representational similarity are a strikingly consistent set: rare, technical nouns and their plurals Figure 4c. We hypothesize that bringing words close to their inflected forms is a proxy effect of learning their meaning, since these forms have nearly identical semantics. By contrast, the pairs that have the greatest *decrease* in similarity are common words like "of" with word fragments like "teasp", as shown in Appendix G, Figure 13. Both of our qualitative findings corroborate our frequency results from Section 5, highlighting frequency as a primary factor in driving embedding changes.

## 7 DISCUSSION AND LIMITATIONS

We have used representational similarity analysis to examine how vocabulary embeddings in language models evolve during training, revealing that semantic and syntactic structures emerge early while frequency effects have continuous influence on the geometry. Later training primarily refines morphological relationships between rare words, showing how vocabulary representations organize around linguistic structure with distinct roles for word frequency and function. Since frequency has such far-reaching effects in the lexicon, a key limitation is that our current experimental paradigm cannot distinguish between the effects of frequency and some of the other factors that we test. Therefore, a promising avenue for future work would be to try and isolate signatures in lexicon training dynamics that are more provably effects *beyond* the role of frequency. One way to do this would be to track the representational changes of each word individually (as opposed to grouped into meaningful clusters), and then tease out the effects of frequency versus other features.

Analyzing and understanding the embedding matrix of LLMs is also an important step in linking LLMs to human language cognition. Firstly, embedding matrices raise an interesting question: does the methodological choice of a large token embedding matrix in LLMs implicitly build in the assumption of a 'words-and-rules' (Pinker, 1998) approach to language into the system? Most LLMs indeed separate these two mechanisms conceptually, between the initial embeddings that represent words, and the subsequent model components that manipulate them. At the same time, studying the training dynamics of embeddings provides a window into how different aspects of language learning may be bootstrapped. We find clear evidence that nontrivial grammatical and semantic structure is emergent within the initial high-dimensional embedding spaces of LLMs, shaping how the lexicon interacts with later layers. The aim of this paper has been to take first steps toward characterizing these lexicon representations: how they encode structural properties of language, and how such organization emerges over training.

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

# A  OLMO RESULTS

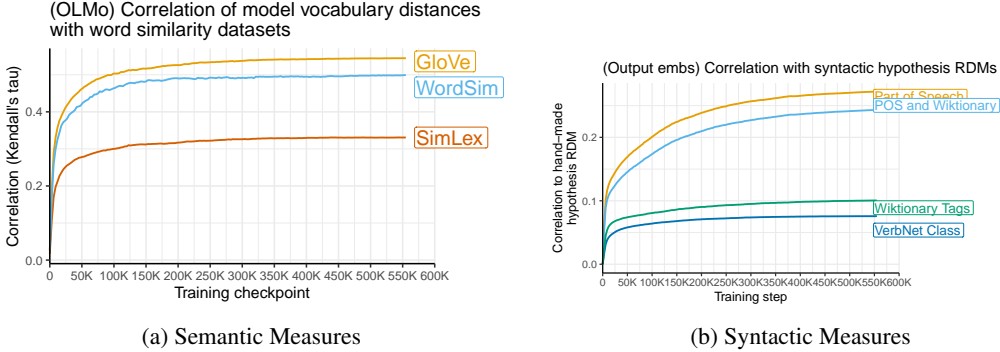

(a) Semantic Measures

(b) Syntactic Measures

Figure 5: OLMo results for Experiment 1

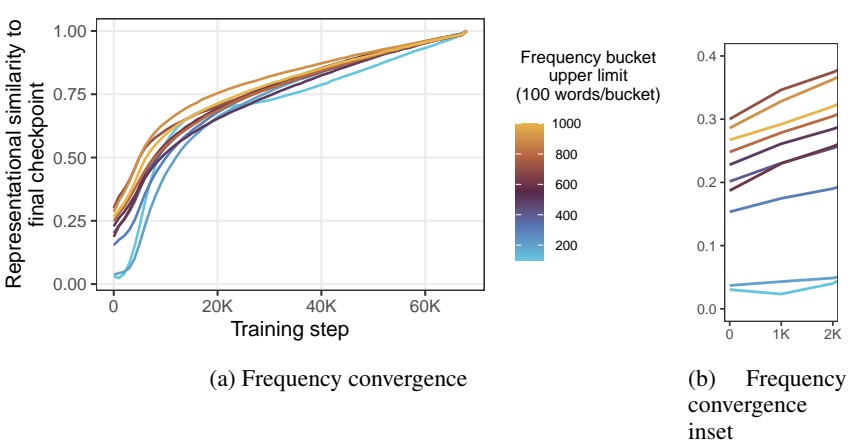

(a) Frequency convergence

(b)  Frequency convergence inset

Figure 6: OLMo results for Experiment 2

# B OUTPUT EMBEDDINGS (UNEMBEDDINGS) RESULTS

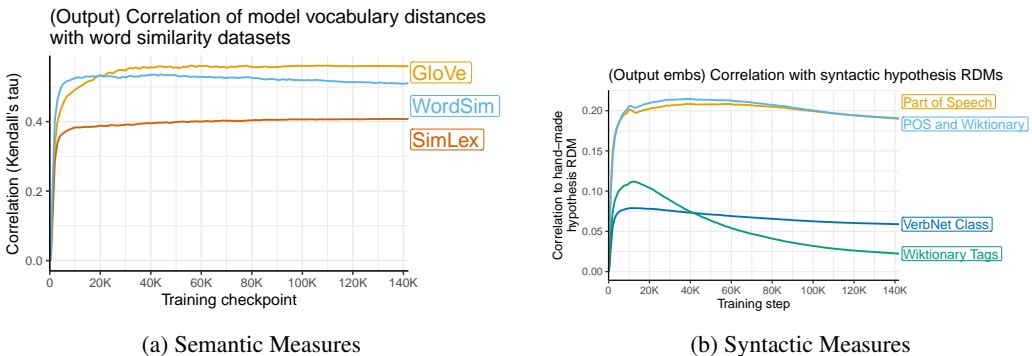

(a) Semantic Measures

(b) Syntactic Measures

Figure 7: Pythia output embeddings results for Experiment 1

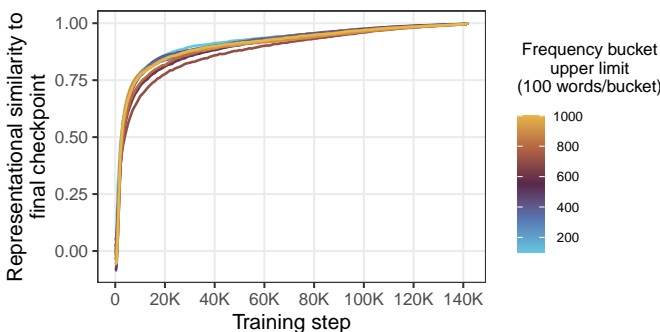

Figure 8: Pythia output embeddings results for Experiment 2

## C  EXPERIMENT 1 WITH COSINE AND EUCLIDEAN DISTANCES

As a sanity check for the effect of using Spearman distance, we re-ran the semantic experiments of Experiment 1 using cosine and Euclidean distances, and found that the results look almost ideantical (Figure 9).

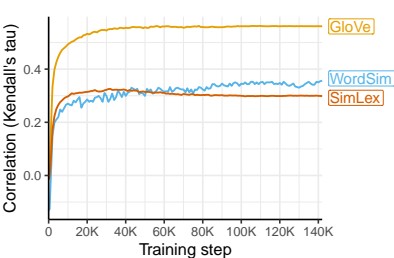

(a) Figure 2b, using cosine distance between token embeddings instead of Spearman

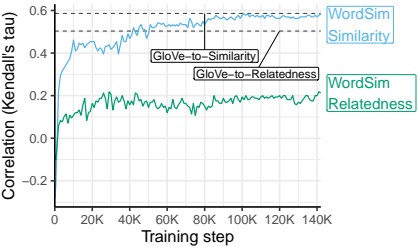

(b) Figure 10 using cosine distance between token embeddings instead of Spearman

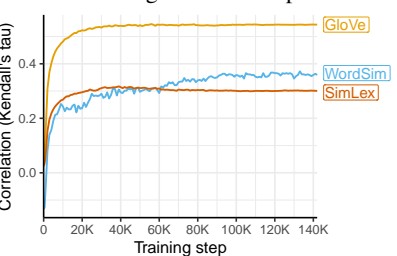

(c) Figure 2b, using Euclidean distance between token embeddings instead of Spearman

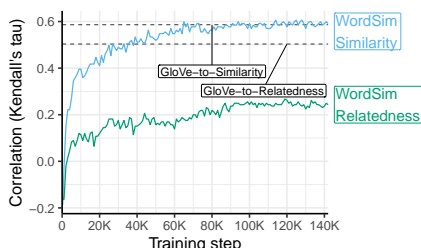

(d) Figure 10 using Euclidean distance between token embeddings instead of Spearman

Figure 9: The semantic correlations of Experiment 1, reproduced using cosine and Euclidean distances in token embedding space as the dissimilarity metric.

# D  SIMILARITY AND RELATEDNESS SPLITS

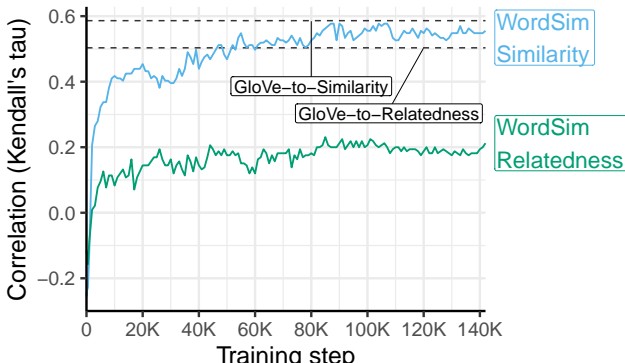

Figure 10: Model embeddings represent similarity much more than they represent relatedness. This is in contrast to our baseline of GloVe vectors (dotted black lines), which correlate with the Similarity and Relatedness splits much more similarly.

# E    REGEX FOR COUNTING WORD FREQUENCY

To determine the frequency of each word, we counted the number of times every string that matched the following regex appeared in the C4 corpus (Raffel et al., 2020).

```
\d+(?:,\d{3})*(?:\.\d+)?|
(?<!-)\w+(?:-\w+){1,3}(?!-)\b|
(?:\w{1,3}\.){2,}\w{0,3}|(?:\w{1,3}\.)+\w{1,3}\b|
\w+(?:[']\w+)*|
\w+
```

This captures:

- Numbers with commas and decimals
- Hyphenated words (well-known, know-it-all), excluding sequences longer than 3 words (which are often websites etc)
- Acronyms (U.S.A., Ph.D)
- Words with apostrophes (don't, can't)
- Standard words

## F  CONVERGENCE RSA FOR SYNTACTIC CATEGORIES

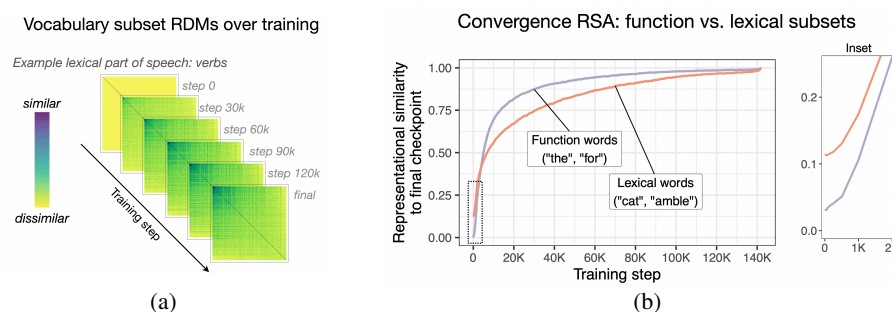

(a)                                    (b)

Figure 11: **Convergence RSA: convergence to final checkpoint by part of speech**. The pairwise distances of function (or, grammatical) words converge to the relationships in the final model checkpoint faster than lexical words do. These results are averaged over the functional and lexical parts of speech. **inset:** At training step 0, where the embedding matrix is just a random initialization, the representation of the embedding matrix has non-zero correlation with the final checkpoint. This is especially the case for lexical words. This indicates that model embedding representations cannot fully move away from the representational structure of the random initialization

**Function words converge earlier in training than lexical words**  In Figure 11b, we present our results for the convergence of different parts of speech. We report the average convergence for the two primary classes of parts of speech: functional parts of speech (pronouns, adpositions, auxiliaries, conjunctions, determiners, numerals, particles, and punctuation) and lexical parts of speech (nouns, verbs, proper nouns, adjectives, and adverbs). This differentiates words that have a largely grammatical use to those that usually impart lexical meaning to sentences. Our results show that functional words, which serve a more grammatical purpose, converge to their final representational space earlier in training than lexical words.

**Lexical word representations are correlated with their random initializations**  Lastly, it is notable to examine the very beginning of our syntactic convergence RSA plots, at training step 0, which we reproduce in more detail in the inset in Figure 11b. These values show how correlated the final relationships represented in the model embeddings are to the random initialization of the embedding matrix. Interestingly, this correlation is not zero for either of the two classes, and is even higher than 0.1 for the lexical classes. This indicates that the random initialization influences how models represent the relationships between words, and that this random influence cannot be overcome even with a lot of training.

## G  FULL RESULTS: WORDS THAT MOVE MOST TO BE CLOSER AND FARTHER FROM EACH OTHER AFTER THE FIRST 20,000 TRAINING STEPS:

```
redshift -- redshifts        exponent -- exponents        models -- model
neutrinos -- neutrino        diodes -- diode              motifs -- motif
galaxy -- galaxies           histograms -- histogram      lattice -- lattices
graphs -- graph              vertex -- vertices           tumours -- tumors
qubit -- qubits              manifold -- manifolds        decay -- decays
photons -- photon            fermions -- fermion          subgroup -- subgroups
nous -- Nous                 detector -- detectors        voltage -- voltages
estimators -- estimator      polynomial -- polynomials    antennas -- antenna
```

Figure 12: Top word pairs that get closer between 20,000 training steps and 142,000 training steps. The vast majority of the words that get closer are inflectional forms of rare, technical words. The maximum change in correlation for any two pairs getting closer is 0.47

```
pentru -- Ã®n        0 -- hydrocar     0 -- unmist       better -- refr    where -- glimp
pentru -- ĂLe        van -- zijn       of -- Gmb         know -- Gmb       need -- weap
ÈĻi -- pentre        0 -- uintptr      them -- Gmb       good -- Gmb       cases -- corrid
sÄĥ -- pentre        A -- Leban        the -- deleter    0 -- teasp        play -- unmist
Ã® -- pentru         of -- teasp       into -- unmist    0 -- advant       " -- Gmb
0 -- errnoErr        set -- Gmb        if -- Gmb         any -- ocks       A -- inhal
```

Figure 13: The top word pairs that get farther between 20,000 training steps and 142,000 training steps. The majority of these pairs are function words with unintelligble or word fragments.

