# OpenReview forum: "Vocabulary embeddings organize linguistic structure early in language model training"
_ICLR.cc/2026/Conference — Submitted to ICLR 2026_

### Official Review · Reviewer_sAQS · 2025-10-30

**Soundness:** 3
**Presentation:** 3
**Contribution:** 2
**Rating:** 4
**Confidence:** 5

**Summary:**

This paper studies how the embedding matrix of an LLM evolves over the course of pre-training. Leveraging two variants of representational similarity analysis this work looks at how linguistic features, and frequency information develop in the Olmo and Pythia families of models. Results show linguistic information converges early on, representations for high frequency words also converge early in training with low frequency words taking much longer. Low frequency words also retain similarity structure with their random initialisation more than their high frequency counterparts.

**Strengths:**

This paper is clearly written, giving the reader a clear sense of their methods and results. The information is well presented and largely well scoped to the scale of a conference paper. The results related to frequency information in particular are interesting, shedding some light on how frequency information affects the training process.

**Weaknesses:**

The two main weaknesses appear to be a limited degree of novelty and a lack of theoretical clarity in the relationships the authors draw to linguistics. In terms of novelty, when during training syntactic information is represented in an LLM is well studied, to a lesser degree effects of frequency information are as well. The authors make clear that much of this work has studied model activations rather than the input embedding matrix. While this may be true the overwhelming majority of computation in a model happens inside of the model. As a result though there is novelty here it seems quite narrow.

Of the results, the results showing lower frequency tokens retain similarity structure with their random initialisation are particularly interesting. However the results in these sections are largely a list of empirical findings with limited analysis or explanatory account - instead offering claims like "frequency [is] a primary factor in driving embedding changes" (a fact established in previous work), the question is how frequency drives those changes. The authors have some results to this effect, it would be great to see a clearer line of argumentation of how frequency affects training.

The stated relationship between this work and linguistics is difficult to follow. Looking particularly at the "Structure in the vocabulary" paragraph (line 112). The "words and rules" approach attributed in Pinker is in fact the broader Generative Tradition in linguistics, which has historically emphasised the separation of syntactic and semantic processes. These are contrasted by Usage-Based approaches to language which argue syntax and semantics are broadly inseparable, with semantic constructions also being a part of the generating process. As written this paper describes this as the "syntax-lexicon" continuum --- a lexicon is not the same as semantics. All accounts cited broadly agree with the existence of syntax and a lexicon, but disagree about what is in the lexicon - this is the line of argumentation that aligns with the authors experiments but as a reader is not the argument they appeared to make. For reference there's a particular disagreement between Jackendoff 1990 and Goldberg 1995 (which they cite) about the degree to which syntactic information is present in the lexicon.

An issue with drawing comparisons with Usage-Based accounts is that they emphasise the importance of context in conditioning and driving the generative process, the author's choice to consider only the embedding matrix makes it hard to directly relate the results here on non-contextualised embeddings to this work in linguistics. Additionally in Usage-Based accounts we would expect semantic, syntactic, and frequency information to be entangled. The analysis here isolates proxies for these kinds of information in separate labels rather than in combination - the authors note this point at 348 that embeddings may be "better explained by non-trivial combinations of linguistic features".

In the conclusion the authors state: "Does the methodological choice of a large token embedding matrix in LLMs implicitly build in the assumption of a words-and-rules approach to language in the system?" I struggle to understand what this question is asking. In the human case, we take raw words, or phonemes as inputs yet may process them as contextualised semantic concepts like constructions. How could having a discrete input, or an embedding matrix for your discrete input entail a factorisation of syntax and semantics?

To be clear I think the methods here are sound, and some of the results are interesting. However the authors emphasise the relationship between their work and related work in linguistics, in part to add theoretical novelty to their use of only the embedding matrix. As a result their engagement with linguistic theory needs to be accurate, and well described.

**Questions:**

How could "the methodological choice of a large token embedding matrix in LLMs implicitly build in the assumption of a words-and-rules approach to language in the system?"

Virtually all kinds of structure you analyse saturate in the first half of pre-training. Are you claiming no further linguistic information is learned later on?

Does when "linguistic features" stabilise relate to performance in any way? Olmo has a higher peak, is it better at a particular task?

---

> ### Author Response · Authors · 2025-11-21
>
> Thanks for your insightful review! It really gets at some of the questions that we have with this paper, and we’re glad to get a chance to expand on this and clear up our argument. We’ll first discuss some of the more general points brought up in your review, and then answer some specific questions.
>
> ### Theoretical clarity
> We really appreciate your comments on theoretical clarity, this is something that we are interested in and had originally spent more text fleshing out. Unfortunately, with the space limitations, and since ICLR is a general ML conference where this type of linguistic theory is not typically what’s most valued, we cut some of this out. Re-reading it now with fresh eyes, we see that you’re definitely right that it ends up a bit muddled and disjointed in its conciseness, and we’ll change this in re-writes, adding back some clarity about how our experiments relate to ideas about the lexicon.
>
> Firstly, there is the issue of **why we are looking at the static embeddings**. This is more readily explained by the transformer architecture rather than by any theoretical linguistic motivation, which we will aim to make more clear in rewriting. The reason to care about the static embeddings is that, on an informational level, any linguistic processing that we observe in a transformer stems from the word embedding vectors. That is, if we have two runs of a language model with two different inputs, the only difference between those two runs is the word embedding vectors, and any further grammatical or semantic processing in the weights (attention weights, MLP weights etc) is somehow indirectly triggered through the information in the vectors. We believe that a serious interpretability research program will include a thorough mechanistic understanding of where information starts from, and how it interacts with model weights to lead to complex linguistic processing over training, and the first goal of this program should be to understand the static embeddings.
>
> Whether this transformer-style approach is a reasonable way to think about the human lexicon and its relationship to grammatical processing is unclear, but it is an interesting model to consider as a functional theory, especially given its behavioral successes in LLMs. Nevertheless, this architectural observation is our motivation for looking at the static embeddings, and our motivation for using RSA: RSA is a measure of abstracted representational similarity, which can tell us about the information encoded in distances. This makes it a good way to get at what we are trying to measure, which is the linguistic information in the embeddings that can be used later on in the model.
>
> The point where we wanted to draw a connection with the background theoretical research is in **what we want to look for in the static embeddings**. The primary connection that we want to draw attention to for the ML world is that the role of the lexicon is a richly debated topic, without a clear answer of what information might be contained in a mental lexicon (if indeed such a notion makes sense to think about). Therefore, in terms of the information-theoretic representational context of the static embeddings, there is a large range of possibilities for the hypotheses of what can be represented. Though it’s simple to think that word meaning and grammatical processing are separated between the static embeddings and the rest of the weights (and we suspect that the implicit belief in a lot of the LLM community), there is no reason to believe that the optimization has settled on this.
>
> ### Novelty
> Regarding novelty, we are not aware of too much work that does these types of analyses, examining a suite of features, and how they develop geometrically and relate to training convergence. Certainly frequency and syntax have been a concern in a lot of interpretability and training dynamics work, but in terms of a detailed account of the geometric training dynamics of language models, we think that our findings definitely add to the overall knowledge in the field.
>
> We find that our results create knowledge which is a necessary (conference-scoped) contribution step for our larger research program, which is to develop a thorough account of how linguistic information is learned over training. This involves understanding how information arises in the word embeddings (this work) and then how the processing of these embeddings co-develops to lead to the linguistically-complex processing that we see in later activations and behavior.

---

> > ### Author Response · Authors · 2025-11-21
> >
> > ### Q1: The theoretical linguistic connection to the transformer architecture.
> > Thanks for pointing this out! What we meant by that line is that, in some sense, the architecture of transformer LMs seems to impose this generativist separation between a lexicon and grammatical processing, neatly separating out weights for the lexicon and other weights that can encode something like a grammar. However, though this is one way to interpret what is being done with LM weights, this is not the only one. The truth is that the continuous optimization space can actually end up on many different structural hypotheses about what information goes into the lexicon  and what the later processing weights do with it.
> >
> > That being said, the current language modeling paradigm does impose certain biases through the architecture, where certain types of information (like, construction information or grammatical forms) are likely harder to encode in the lexicon, since it is already partitioned, so the vectors must in some sense represent those tokens that it is given. The pre-partitioning of the lexicon by the tokenizer, before any other language learning, is one of the key ways in which the language modeling paradigm strays from cognitive plausibility. In our discussion, we wanted to point to the ways in which the architectural choices relate to and bias the processing to conform certain ideas about language, but without necessarily restricting the processing to any one easily-formulated idea, since the optimization can reach many complex solutions.
> >
> >
> > ### Q2 What is learned later on
> > What our findings show is that a lot of the linguistic information in the word embeddings stabilizes early in training (or, interestingly, peaks). There are multiple things that can happen after this, however: the rest of the weights can continue to learn how to use these features, and the embeddings can change in ways not captured by our metrics, which we look at in Experiment 3, and see that. In Experiment 3, we try to address this latter question, and see that the embedding representation space does keep changing after feature stabilization (Figure 4a) and that the thing that is most changing in embeddings is learning the semantics of long-tail low frequency words (Figure 4c, complete in appendix Figure 12)
> >
> > ### Q3 Relation to linguistic performance
> > The difference between the two models is not likely to be behavioral, but in their processing mechanisms (to make sure, we want to verify this first claim, but SyntaxGym seems to not be reachable so we are setting up some syntactic testing that will take us a bit longer.) It is unlikely that OLMo knows more about part of speech than Pythia, which is a 12B-parameter language model that is going to have a very good sense of part of speech. However, it seems that OLMo stores this information much more accessibly in the word embeddings, which likely signals a different mechanism of processing that as we state in the paper we are very interested in looking into.

---

### Official Review · Reviewer_UGW7 · 2025-10-31

**Soundness:** 2
**Presentation:** 2
**Contribution:** 2
**Rating:** 2
**Confidence:** 3

**Summary:**

This paper studies how and when the (input) vocabulary embeddings of Large Language Models (LLMs) organize themselves to reflect linguistic properties. Using Representational Similarity Analysis (RSA) on the Pythia 12B and OLMo 7B models, the authors track the pairwise distances of the input embeddings throughout the training process. They find that this linguistic structure emerges very early in training, since the embedding space quickly converges to a state that shows high correlation with both semantic features and syntactic features. They also find that word frequency plays a major role as high-frequency words converge to their final representations much faster than low-frequency words.

**Strengths:**

This study has a strong empirical scope as it assembles dense checkpoint trajectories across two modern model families. The applied RSA technique is principled, and the reported early semantic plateau is clear and replicated across models. The frequency-aware slicing idea and the comparison to the final checkpoint are interesting.

**Weaknesses:**

**W1. Operational significance is unclear.**

The paper reports Hypothesis-Driven RSA and Convergence RSA curves, but it does not show how their curves relate to model perplexity or to any simple downstream behavior. Because perplexity is the training objective, it is important to see whether the reported “early emergence” (for semantics) or “early peak” (for syntax) coincides with changes in perplexity. Without such a link, the reader cannot tell whether these RSA events matter for model quality or are just descriptive patterns.

**W2. Frequency is a likely confound that is not controlled.**

The paper reports "early" emergence for both semantic and syntactic RSA (Section 4), but it never controls for token exposure. Frequent words appear many more times and therefore get many more updates; this alone can create an "early" effect. Section 5 acknowledges the role of frequency by slicing into buckets, yet it does not revisit the "syntax" claim after holding frequency fixed. As written, "early syntax" can simply be a frequency artifact.
To resolve this concern, the authors should modify their Experiment 1 to build matched sets with controlled frequency distributions and recompute the RSA curves.

**W3. Cross-bucket (global) structure is not measured.**

Section 5 looks only within each bucket (100×100 distance matrices per bucket). It never examines how different buckets sit relative to each other over time (for example, whether the center of the top-100 words moves closer to or farther from the center of the 900–1000 group). It is entirely possible that the global arrangement across buckets stabilizes early while the finer within-bucket neighborhoods keep shuffling. If so, the paper's timing claims would be incomplete.

**W4. "Convergence to final" is fragile and can mislead**

Convergence RSA defines "stability" as "becoming similar to the final checkpoint." This assumes the last snapshot is the right anchor, even though late steps can have extra noise or idiosyncrasies. As a result, a subset can look "stable" simply because it mirrors quirks of the final checkpoint. This matters because the paper's main takeaways about "who stabilizes when" depend on this anchor. It also affects the interpretation in Section 5: when time is rescaled by expected exposures, the narrative ("low-frequency words converge quickly per exposure") competes with another observation in the paper, i.e., low-frequency words keep stronger ties to initialization. If those words have not actually stabilized by the final checkpoint, "convergence to final" is not a reliable guide for them.

**W5. Claims and presentation need correction and calibration.**

The paper uses ambiguous terminology and legends. Two examples are given below:

- In Section 6, the text alternates among "distance decreased," "embeddings moved closer," and "similarity increased" to describe the same pairwise change. This slows the reader and risks confusion.

- In Figure 3(a), the legend says "Frequency bucket upper limit (100 words/bucket)," but the numbers are frequency ranks (larger number = less frequent), so the correct label is "Frequency rank bucket (100 words/bucket)".

Furthermore, some of the statements in the paper seem overconfident. In Figure 4(b), the text states that "high-frequency words (light blue) show substantially stronger correlations between input and output embeddings than low-frequency words," yet several lower-frequency curves sit above higher-frequency ones at many points, and no uncertainty bands are shown. The reasonable claim here is "on average" rather than an unconditional statement.

**Questions:**

Q1: What do the reported RSA patterns (your checkpoint-wise correlations between embedding distances and linguistic hypotheses) tell us about model quality in practice? Do the early emergence or early peak you highlight correspond to any observable improvement in the model’s ability to predict text or handle simple behaviors?

Q2: Please clarify the contribution to interpretability beyond descriptive timelines. Do these observations translate into explicit actions in practice, and if so, which training or evaluation decisions should reasonably be guided by them?

Q3: In Section 4, how do you distinguish structure that arises because certain words appear far more often during training from structure that reflects genuine syntactic or semantic organization?

Q4: In Section 5, how does the relationship between frequency groups evolve (e.g., head vs. tail as a whole)? Do cross-group relationships stabilize earlier, later, or differently than within-group neighborhoods?

Q5: In Section 5, why is the last checkpoint an appropriate reference for your "convergence" analyses? How sensitive are your conclusions to that choice, and are low-frequency words actually stable by the end of training?

Q6: Several frequency buckets appear out of order in figures (e.g., a lower-frequency curve above a higher-frequency one). How should readers reconcile this with statements that high-frequency words show "substantially stronger" effects?

Q7: Related to Q6, what level of variability or uncertainty is present in your curves and bucket comparisons? Can you clarify how confident we should be about the ordering across frequency groups?

---

> ### Author Response · Authors · 2025-11-28
>
> ### W1: Operational significance
> * This is a reasonable suggestion for future work. To clarify the scope and aims of our contribution: the specific aim here is to understand when and how linguistic structure emerges within the embedding space that all downstream computations depend on. We argue that this is a valuable representational study in its own right, much like how neuroscientists may publish studies of brain representation without simultaneously proposing a new clinical intervention.
> * That said, the relationship between this embedding geometry and model perplexity is likely meaningful, with stabilization of semantic/syntactic correlations roughly coinciding with a period of steep perplexity decline in both Pythia and OLMo training curves. Establishing a causal or predictive link would require controlled experiments that are beyond the scope of this paper, but the reviewer is likely correct in their intuition that these RSA events bear direct relationship to gains in model capacity, and we would be eager to explore this in future work.
>
> ### W2 & Q3 Frequency as an uncontrolled confound?
> * We appreciate that the reviewer raised this point. Experiment 2 explicitly digs into frequency effects, showing systematic differences in convergence as a function of frequency buckets. The reviewer’s suggestion to “revisit the syntax claim” is tricky in practice since frequency and syntactic category are naturally entangled, with function words tending to be more high frequency, and content words are all over the distribution. Critically, our finding that syntactic correlations peak and then decline (Figure 2c) is hard to square with a pure frequency story. If this were just about frequent tokens accumulating more updates, we'd expect the correlation to keep climbing monotonically, not hit a peak and level off or drop. That non-monotonic pattern suggests something beyond frequency is going on, with the early emergence of syntactic information.
>
> ### W3 & Q4: Measuring cross-bucket frequency structure
> * This is a fair point and an interesting question for further analysis. We will consider adding this analysis if space permits, though we note it is complementary to rather than invalidating of our current findings.
> * In general we are not convinced that cross-bucket analysis is the ideal level of analysis for understanding linguistic structure in embeddings. Semantic and syntactic organization don't naturally respect frequency boundaries: a rare noun like "quark" should have an embedding that is close to other physics terms regardless of their frequency, not clustered with other rare words. Our within-bucket analysis captures whether words are finding their linguistically appropriate neighbors, while a cross-bucket analysis would mostly reflect frequency stratification in the embedding space.
> * If the reviewer has a specific hypothesis about what cross-bucket dynamics would reveal that our within-bucket analysis misses, we’d welcome if they shared that idea with us in their response.
>
> ### W4 & Q5 The “convergence to final” measure
> * The reviewer raises various concerns about our use of the final training checkpoint as a key target for comparison in the convergence RSA analyses. There are two main reasons we did so: (1) it is the checkpoint that would be deployed for use and analysis by the field, making it practically relevant, and (2) it provided a consistent reference point across all analyses.  We feel that our comparisons of representational geometry rather than raw distances somewhat mitigates the reviewer’s concern, as relational structure between token embeddings is more likely to be robust to idiosyncratic noise/changes that occur during the late stages of training, as opposed to raw embedding distances. Moreover, the highly consistent patterns we find between Pythia and OLMo suggest that our results aren’t overly influenced by checkpoint-specific artifacts.
>
> W5: Terminology around certain claims
> * We thank the reviewer for these helpful critiques. We will standardize the terminology around pairwise changes, correct the Figure 3a legend, and add appropriate hedging to certain claims regarding the frequency bucket comparisons (e.g. “on average”), as suggested.

---

> > ### Author Response · Authors · 2025-11-28
> >
> > ### Q1: Relationship with model performance
> > The peaks that we show happen during a rapid fall in perplexity in the models’ loss curves, during early training. The perplexity continues dropping after this (rather than fully evening out, as our curves often do), as the model continues learning. Therefore, it’s fair to say that our metrics reflect some of the features that make model perplexity drop, but not all of them.
> >
> >
> > ### Q2 Training recommendations from these findings
> > We envision that these results form the beginning of a research program that mechanistically understands training: how linguistic information is learned over training, and how the processing of linguistic information throughout the layers is acquired. Such a research program would naturally start by understanding the training dynamics of the input embeddings. As with a lot of basic science interpretability work, there isn’t a clear path to suggesting model improvements from the results, as the main goal is to understand how current models do what they do
> >
> > ### Q6: Why aren’t frequency effects monotonic?
> > Correlation with frequency (rather than frequency rank) is non-monotonic, while correlations with frequency rank are monotonic. This means that the vocabulary first starts off reflecting the huge differences in frequency count and then does so less, while maintaining a general structure that words that are of similar frequency ranks have similar geometries. Due to the zipfian nature of vocabulary frequency, the counts of the frequent words are huge and distances are massive, while in frequency rank, the distances between each consecutive word are all equal. What the pattern that we see means is most likely that the very frequent function words move together earlier (when most words have not moved) much more so than any other words, but then as most words are seen and start moving, the frequency starts being a more equally-spaced measure in how it influences direction.

---

### Official Review · Reviewer_y9Ge · 2025-10-31

**Soundness:** 2
**Presentation:** 1
**Contribution:** 2
**Rating:** 4
**Confidence:** 4

**Summary:**

This work uses representational similarity analysis (RSA) to examine how vocabulary embeddings in language models evolve during the training phase. Basically, they conduct experiments to study the correlation between embedding-based distance and human-annotated distance for several tasks: semantic similarity, syntactic similarity, and word frequency. Their main findings indicate that
* The semantic and syntactic converge to high correlations at an early training step
* Embeddings of high-frequency and function words converge to their final vectors faster than lexical and low-frequency words

**Strengths:**

* This work explores how does he embedding layer in transformers learn knowledge durting the training stage, which is interesting and novel
* Their findings track the representative changes of tokens and reveal their connections between initial random embeddings and the learned embeddings. The study of the less-frequent tokens might be useful for future studies since LLMs tend to fail on long-tail and domain-specific cases.

**Weaknesses:**

* I appreciate that the authors aim to answer this question: what are the representative changes of word embeddings during training, and this work provides some findings. However, I would expect some more insightful analyses. (1) LLMs are known for their contextual capabilities, which means they understand each word according to the context surrounding it rather than using the static embedding layer. What is the motivation to study this static embedding matrices? Can we apply the main findings in this manuscript to improve the current architecture or training strategies? (2) the authors found that the semantic and syntactic converge to high correlations at an early training step. Why do the embeddings exhibit such a pattern?  It would be more helpful if the authors could go further and offer explanation for their observations. (3)  The second main finding *"Embeddings of high-frequency and function words converge to their final vectors faster than lexical and low-frequency words"* is aligned with our intuition, which is not surprising. I would suggest exploring this point: one known issue of LLMs is that they tend to generate hallucinated answers for long-tail queries. Is it possible to learn the correlation between hallucination and low-frequency words? If so, it would be very useful to find solutions to mitigate this issue.
* The presentation requires another round of revision. I have difficulties understanding the details of the methodology and I have to guess to complete this review. I would suggest adding basic notations for clarification.

**Questions:**

* It is not clear how the authors perform the hypothesis RSA. According to my understanding, the task aims to assess the correlation between embedding‐based distances and human-annotated labels, but each step in this process is not clearly stated. It would be useful to introduce some notations to describe the problem, input, output and evaluation protocol. I have difficulties understanding this RSA, especially for the syntactic analysis.
* In line 278 *"and if a word appears as a part of speech at least 5 times it receives distance 0 to all other words with that part of speech"*, I do not fully understand what is the meaning of this sentence

---

> ### Author Response · Authors · 2025-11-21
>
> Thanks for your review! We address the questions you raise below, thanks for your constructive comments.
>
> ### W1: Why are we just looking at word embeddings?
>  You are definitely right that language models are set apart from previous NLP methods in that they utilize context, and most of recent interpretability research has focused on looking at embeddings in context. However, when it comes to linguistic features, the question is not so much *whether* they are there in the contextual embeddings, but a general mechanistic one: where does this contextual linguistic information come from and how is it built over training and over layers? To start to tackle these questions, it is important to understand the word embeddings, which is **where all contextual information starts to be built from**
>
> The static word embeddings are important in understanding how information is built because they are the *differentiation weights* of a model. When we give two different inputs to the same model, we get very different processing. This all stems from the input word embeddings: the other weights that are mobilized (the attention, the  MLP) are identical between the two runs. Therefore, from a training dynamics perspective, it is important to understand when this information, that stems from the word embeddings, enters the learning system during training.
>
> We envision that these results form the beginning of a research program that mechanistically understands training: how linguistic information is learned over training, and how the processing of linguistic information throughout the layers is acquired. Such a research program would invariably start by understanding the training dynamics of the input embeddings. As with a lot of basic science interpretability work, there isn’t a clear path to suggesting model improvements from the results, as the main goal is to understand how current models do what they do.  We’ll be sure to make all of these points clear in the rewrite.
>
> ### W2: Explanations for early convergence.
>  The paper is aiming to separate two things that happen during training: 1) information is learned and put in the word embeddings (as we said previously, essentially all of the information that is used by a language model for a specific input is in some way inputted through the word embeddings, as all the other weights are the same for every input), and 2) the rest of the weights learn to process this information in complex ways.
> What the early point of convergence tells us is that the first type of information is learned quickly, with the exception of low-frequency tokens. It is not a priori obvious whether this information would be learned early or co-evolving with the rest of training, and this is what we show. The explanations for this could be many, but overall it is an intuitive learning paradigm, with lexical information being made solid and then used, which we showed that LLMs follow.
>
> ### W3 Frequency results and long-tail hallucination
> You are certainly right, that it is intuitive that words that are seen more often converge more quickly, though as far as we are aware it hasn’t been shown. What is interesting in our case is the analyses around this fact. Firstly, we show that frequent words converge a lot slower *per exposure* than less frequent words (Figure 3a, right side). Secondly, we show that less frequent words remain correlated to the distance relation biases in their random initialization. Our RSA methodology is uniquely suited to show this representational similarity between the random initialization and the final embeddings.
>
> Low-frequency vocabulary items are certainly a possibility for how hallucinations come about. As we show in our qualitative results (Figure 4c), low-frequency items are the last to get situated in intuitive semantic distance relationships with other words in the embedding space, and perhaps not all do.
>
> ### Presentation
> Thanks for pointing this out, re-reading  we see that we introduce some of the notation in Section 2.2, and then use it in the methods section (3) without re-defining. With the extra space, we will take care to explain the methods better, flesh out the caption of the method diagram, and use less in-line notation in order to make the presentation easier to follow.
>
> ### Question about ln 278
> With this line, we are explaining how we make the part-of-speech hypothesis matrix. We take every word, look through UD, and see which parts of speech it takes. For example, the word “run” is likely to appear many times as a verb and a few times as a noun. Then, for every part of speech, we take every word that takes that part of speech at least 5 times (to account for noise or mislabellings or strange edge cases), and make all of those word-word distances 0. We show our POS matrix in Figure 2a. You’re right that that line is quite confusing in its conciseness, we’ll change that in the rewrite, and read through for other things like this.

---

### Official Review · Reviewer_SqcY · 2025-10-31

**Soundness:** 4
**Presentation:** 2
**Contribution:** 3
**Rating:** 6
**Confidence:** 3

**Summary:**

The paper is a descriptive study that investigates how vocabulary representations of language models evolve during training. The experiments focus on two English language models (Pythia-12B and OLMo-7B) for which several training checkpoints are available, allowing to study their evolution over time.

Across three experiments, the paper studies the relationship between the vocabulary representations at each training checkpoint and 1) human annotated datasets for vocabulary structure (semantic and syntactic similarity), 2) word frequency in the training corpus and 3) the final representation of the fully trained model.

The paper uses Representational Similarity Analysis (RSA) to compare the proximity between two different representational spaces, model's representations and the human's annotation (or word frequency).

The study finds that the models' representations of wordss correlate most strongly with human semantic and syntactic annotations early in training (around 15% of total steps), that representations of frequent words converge earlier than those of less frequent words, but also that nevertheless, the embeddings continue to change during the remainder of training, especially for rare and technical tokens.

The paper claims that these findings could help to better understand the training mechanisms of language models.

**Strengths:**

1) The paper provides a handful of experiments to characterize vocabulary representation across several dimensions. The experiments and claims are clearly explained. I found the combination of hypothesis RSA and convergence RSA especially interesting.

2) This methodology could be adapted to other languages and very interesting, especially for language with structures highly different from English, though this would require significant additional resources.

**Weaknesses:**

1) The paper studies two models, but doesn't and explicitely states that those models are in English and that the claims are only supported for that language. I also think that more models might be necessary to confirm the claims. The paper would benefit from quickly clarifying their architectures, and the content of their training data.

2) Figure are sometimes distant from their discussion in the text (figure 2, and figures in the appendix). The main figures appear to combine both models to show general trends, while the details for each model are in appendix.
To me at least some of the per-model results contain key information that should appear in the main body. Having to switch repeatedly between the text and appendix made the results harder to follow.

3) The experiments are thorough, but some findings (faster convergence of frequent tokens) are largely intuitive, which limits the novelty of the contribution

**Questions:**

1) If I understood correctly, convergence RSA compares the model's representation at different training stages. While RSA seems suited for comparing model representations to human annotated ressources, is it equally appropriate when the representational system is the same? Can other comparisons could be considered in that case?

2) Given the relatively small number of word pairs in Experiment 1, how representative is this subset of the vocabulary, and could this affect the reported trends?

---

> ### Author Response · Authors · 2025-11-21
>
> Thanks for your encouraging review! We address the issues and questions that you bring up below, thanks again for your constructive comments
>
> ### W1 Two models, only in English
> Thanks for this comment, we’ll make sure to make it more clear that our analyses are English-based and the implications of this! We currently mention that our analyses are based only on the English lexicon explicitly in the second paragraph of the paper (lines 45-48), but we’ll also add that the models are trained primarily on English text, and mention the scientific limitations that can come with monolingual analyses.
>
> To run our analyses, we need to have the full training checkpoints of a large model over training. This also means that each of our analyses are run on hundreds of model checkpoints, which is fairly expensive on an academic budget. There aren’t many models like Pythia and OLMo that release all of their training checkpoints and training recipe, and so these two provide an broader picture while still being feasible.
>
> ### W2 Figures far away from text
> With the extra space we’ll definitely include the OLMo results in the main text, and move things around a bit better now that we’re not as space constrained to try and make the overall flow and presentation of the paper better
>
> ### W3 Results are not very surprising
> We think that our results create knowledge which is a necessary (conference-scoped) contribution step in our larger research program, which is to develop a thorough account of how linguistic information is learned over training. This involves understanding how information arises in the word embeddings (this work) and then how the processing of these embeddings co-develops to lead to the linguistically-complex processing that we see in later activations and behavior.
>
> Regarding novelty, we are not aware of too much work that does these types of analyses, examining a suite of features, and how they develop geometrically and relate to training convergence. Certainly frequency and syntax have been a concern in a lot of interpretability and training dynamics work, it would be quite odd if this were not the case. But in terms of a detailed account of the training dynamics of language models, and how the geometric development connects to feature annotations and different facets of frequency, we think that our findings definitely add to the overall knowledge in the field.
>
> Thanks for this comment, we’ll try to make the necessity/novelty of the work more clear in the discussion of future rewrites.
>
> ### Q1 Is RSA appropriate for the convergence case?
>
> You’re right, it’s definitely the case that RSA is much more a necessity in the cases where the representational spaces are totally different (the Hypothesis RSA case) than in cases where we are comparing spaces that are comparable to each other. However, this does not mean that it does not give some valuable insights in the convergence experiments, compared to for example using raw distances.
>
> Firstly, it means that we don’t have to worry about how the space has moved overall, for example, if there is some sort of mostly global drift that we have to take out of raw distances. Of course, it is possible for example to do some procrustean or similar analysis, and try to one of the spaces by some rotation or linear transformation, but using RSA takes out all such needs. Using RSA also lets us have insights that are much harder to see with other methods, like for example that final embeddings are correlated to their initialization biases for less-frequent words.
>
> ### Q2 Are the human-annotated semantic datasets a biased sample
> This is a spot on observation, this is definitely a factor in these experiments and something that we will try to mention more clearly in rewrites. The pairs in the human-annotated biases seem to be biased, in our anecdotal experience, towards pairs of words that (in some sense) it is reasonable to ask about their semantic similarity, which is a bias that is hard to pin down. The word-choosing methodology, as far as we understand, is not standardized or randomized for any of the datasets. SimLex has tried to deal with some of the biases (like concreteness) but not others. The words are also largely higher frequency, which is partly what led us to do the frequency experiments.

---

### Meta-Review · Area_Chair_ipfy · 2026-01-01

**Summary:**

This work evaluates the vocabulary embeddings generated by LLMs by computing the correlation of input embeddings and output embeddings. There is no methodological innovations, just a different way of evaluating existing models. Explanation of the experimental results are largely intuitive, not theoretically justified.

**Reviewer Concerns:**

Most reviewers expressed concern that the nature of this paper is evaluating existing models with limited innovation. Some other concerns include the fact that only two models are considered and the lack of investigating known issues in LLMs such as hallucination.

**Reviewer Scores:**

None is likely to change.

---

### Decision · Program_Chairs · 2026-01-26

Reject